# Targeted Delivery of Butyrate Improves Glucose Homeostasis, Reduces Hepatic Lipid Accumulation and Inflammation in db/db Mice

**DOI:** 10.3390/ijms24054533

**Published:** 2023-02-25

**Authors:** Signe Schultz Pedersen, Michala Prause, Christina Sørensen, Joachim Størling, Thomas Moritz, Eliana Mariño, Nils Billestrup

**Affiliations:** 1Department of Biomedical Sciences, Faculty of Health and Medical Sciences, University of Copenhagen, 2200 København, Denmark; 2Steno Diabetes Center Copenhagen, 2730 Herlev, Denmark; 3Novo Nordisk Foundation Center for Basic Metabolic Research, Faculty of Health and Medical Sciences, University of Copenhagen, 2200 København, Denmark; 4Infection and Immunity Program, Biomedicine Discovery Institute, Department of Biochemistry, Monash University, Melbourne, VIC 3800, Australia

**Keywords:** Type 2 diabetes, butyrate, short chain fatty acids, db/db mice, insulin sensitivity, glucose homeostasis, inflammation, hepatic steatosis, beta-cell function, pancreatic islets

## Abstract

Butyrate produced by the gut microbiota has beneficial effects on metabolism and inflammation. Butyrate-producing bacteria are supported by diets with a high fiber content, such as high-amylose maize starch (HAMS). We investigated the effects of HAMS- and butyrylated HAMS (HAMSB)-supplemented diets on glucose metabolism and inflammation in diabetic db/db mice. Mice fed HAMSB had 8-fold higher fecal butyrate concentration compared to control diet-fed mice. Weekly analysis of fasting blood glucose showed a significant reduction in HAMSB-fed mice when the area under the curve for all five weeks was analyzed. Following treatment, fasting glucose and insulin analysis showed increased homeostatic model assessment (HOMA) insulin sensitivity in the HAMSB-fed mice. Glucose-stimulated insulin release from isolated islets did not differ between the groups, while insulin content was increased by 36% in islets of the HAMSB-fed mice. Expression of *insulin 2* was also significantly increased in islets of the HAMSB-fed mice, while no difference in expression of *insulin 1*, *pancreatic and duodenal homeobox 1*, *MAF bZIP transcription factor A* and *urocortin 3* between the groups was observed. Hepatic triglycerides in the livers of the HAMSB-fed mice were significantly reduced. Finally, mRNA markers of inflammation in liver and adipose tissue were reduced in mice fed HAMSB. These findings suggest that HAMSB-supplemented diet improves glucose metabolism in the db/db mice, and reduces inflammation in insulin-sensitive tissues.

## 1. Introduction

Type 2 diabetes (T2D) is a metabolic disease, characterized by hyperglycemia, that affects over 450 million people worldwide [1]. Both genetic and environmental factors play a role in the etiology, but the rapid rise in the incidence in the last decades is likely explained by lifestyle changes [2]. Physical inactivity and calorie-dense diets increase the risk of obesity associated with metabolic complications, such as insulin resistance and low-grade inflammation [3]. Insulin resistance increases the demand of pancreatic beta-cells to secrete insulin and, in susceptible individuals, the beta-cells eventually fail to produce and secrete sufficient amount of insulin to maintain normoglycemia [4]. Hyperglycemia causes serious life-threating, long-term complications, including cardiovascular disease and kidney failure, and therefore, strategies to prevent T2D are of great interest [2].

Growing evidence shows that the gut microbiota plays an important role in health and metabolic diseases. The gut microbiota produces short-chain fatty acids (SCFAs), such as acetate, propionate and butyrate, through fermentation of dietary fibers [5]. These SCFAs are immune-regulatory, promote gut health and play an important role in host homeostasis, including the maintenance of a healthy gut microbiota critical to control metabolic diseases [6]. SCFAs exert functions in the gut, but they also enter the circulation and reach peripheral tissues and organs [7,8,9]. Individuals with T2D have consistently shown a reduced abundance of butyrate-producing bacteria in the gut [10,11,12,13,14]. Furthermore, a direct correlation between beta-cell function and fecal butyrate concentrations has been found, indicating a role for butyrate in the regulation of glucose metabolism [12]. Therefore, targeting the gut microbiota and the production of SCFAs may have potential in the prevention or treatment of T2D.

Several studies have shown positive effects of butyrate on obesity, inflammation, insulin resistance, and liver dysfunction in rodent models, with various mechanisms identified [15,16,17,18,19,20,21,22]. However, since conventional routes of administrating butyrate do not model a sustained release and absorption of butyrate in the colon and into the circulation, clinical translation may be limited [23]. Moreover, butyrate has a short half-life (15–30 min) in circulation, and is found in low concentrations compared to acetate and propionate [24,25]. For example, rectal administration of butyrate in obese men only resulted in a transient increase in plasma butyrate [26]. This shows that frequent delivery is necessary for sustained release as only limited effects of oral butyrate in individuals with T2D [27,28] or type 1 diabetes (T1D) [29,30] have been reported.

An alternative approach for sustained delivery of SCFAs to the gut and into the circulation is to modulate the gut microbiota to promote a constant production of SCFAs. For example, a specially designed prebiotic starch (from high-amylose maize starch, HAMS) can be used as a carrier for butyrate (HAMSB) [31]. The starch carrier is highly resistant to digestion by host enzymes, and it specifically delivers butyrate to the colon, where it is liberated by bacterial enzymes. The starch residues themselves are degraded and fermented to SCFAs, thereby also promoting the growth of SCFA-producing bacteria. This is an efficient approach to enhance luminal concentrations of SCFAs in both animals [7,32,33,34] and humans [8,35,36]. In non-obese diabetic (NOD) mice, HAMS bound to both butyrate and acetate (HAMSAB) protected against T1D [7]. Furthermore, immune modulatory effects have been reported in humans with T1D [8] and hypertension [37]. We have previously shown that butyrate prevents cytokine-induced beta-cell dysfunction and inhibits proinflammatory gene expression in vitro [38,39]. Therefore, we hypothesized that butyrate has beneficial effects on beta-cell function and glucose homeostasis in mice prone to T2D. We chose the widely used leptin receptor knockout (db/db) mouse model, in which the lack of leptin signaling causes persistent hyperphagia, obesity, beta-cell dysfunction, and insulin resistance and diabetes [40]. We investigated the effect of butyrate, delivered in the form of HAMSB, on glucose metabolism, beta-cell function, hepatic lipid accumulation and inflammation.

## 2. Results

### 2.1. Butyrate Improves the Function of Islets in db/db Mice In Vitro

We previously showed that butyrate protects pancreatic islets from cytokine-induced dysfunction, and reduces inflammatory NF-κB signaling [38,39]. To determine whether butyrate also exerts protective effects on islets from diabetic animals at different disease stages, islets were isolated from db/db mice at 9 and 13 weeks of age. Glucose-stimulated insulin secretion was measured and compared to islets isolated from 11-week-old control C57BL/6 mice (Figure 1A).

Glucose-stimulated insulin secretion was significantly reduced in the islets of db/db mice at both 9 and 13 weeks of age compared to controls (Figure 1A). Moreover, the islets of db/db mice contained significantly less insulin (Figure 1B). To investigate whether butyrate could improve db/db islet function, islets from the db/db mice were cultured for 10 days, with and without butyrate. Butyrate significantly increased glucose-stimulated insulin secretion compared to islets cultured without butyrate isolated from 13-week-old mice (Figure 1C). Although not statistically significant, we observed a similar increase in glucose-stimulated insulin secretion from islets of 9-week-old mice (Figure 1C). Butyrate had no effect on db/db islet insulin content (Figure 1D). Together, these results indicate that butyrate improves the impaired glucose-stimulated insulin secretion seen in the islets of db/db mice.

### 2.2. Effect of HAMSB on SCFA Production

We next investigated the effect of butyrate on glucose homeostasis in db/db mice. Db/db mice were fed either (1) a control diet (Ctr), (2) a diet supplemented with HAMS or (3) a diet supplemented with HAMSB for 5 weeks (Figure 2A). The HAMS diet was included to differentiate between the effects of butyrate and the increased SCFA production through fermentation of HAMS. After 5 weeks, butyrate concentrations in feces from HAMSB-fed mice were 8-fold higher compared to the levels found in feces from Ctr or HAMS-fed mice (Figure 2B). The HAMS diet increased fecal butyrate 2.5-fold compared to the control diet, although this effect was not statistically significant (Figure 2B). In addition, both HAMS and HAMSB increased fecal propionate (Figure 2C), while fecal acetate was only significantly increased in HAMSB-fed mice (Figure 2D). Total SCFA levels in feces were significantly higher in both HAMS- and HAMSB-fed mice compared to the Ctr group (Figure 2E). We did not observe any significant difference in plasma butyrate (Figure 2F) or acetate (Figure 2H) between the three groups. HAMS-fed mice had increased plasma propionate compared to the Ctr mice (Figure 2G). Total plasma SCFA levels were not different between the groups (Figure 2I). Together, this suggests that HAMSB effectively delivers butyrate, as well as propionate and acetate, to the large intestine, and that HAMS also increases SCFA production.

### 2.3. Effect of HAMSB on Body Weight and Glycemic Control

To follow disease progression, body weight (Figure 3) and fasting blood glucose levels (Figure 4) were assessed weekly. At baseline, total body weight and non-fasting blood glucose did not differ between the three groups (Appendix A). However, after 5 weeks HAMSB-fed mice had gained 15.8 ± 1.6 g of total body mass, whereas the Ctr mice only gained 13.7 ± 1.8 g (Figure 3B). Mice fed HAMS gained 14.9 ± 2.3 g (Figure 3B). The weight differences could not be explained by differences in food consumption, as the average food intake was similar among the groups (Figure 3C).

Over the course of 5 weeks, no major differences in fasting blood glucose were observed between the three groups (Figure 4A). Only in week 3 the HAMSB-fed mice had significantly lower blood glucose levels compared to the Ctr group (Figure 4B). As a measure of blood glucose control during the entire experimental period, we calculated the area under the blood glucose curve. The HAMSB-fed mice had significantly lower blood glucose compared to the Ctr mice (Figure 4B). Furthermore, we observed a non-significant (*p* = 0.059) decrease in plasma insulin in mice fed HAMSB for 5 weeks (Figure 4D). Analysis of fasting blood glucose (Figure 4C) and insulin levels (Figure 4D) together, in the homeostatic model assessment for insulin resistance (HOMA-IR), showed that HAMSB-fed mice had a significantly lower HOMA-IR compared to Ctr mice (Figure 4E). HAMS did not affect blood glucose, plasma insulin levels or HOMA-IR (Figure 4). Together, this indicates that although mice fed HAMSB gained more weight compared to Ctr mice, they developed less severe hyperglycemia and have improved insulin sensitivity. 

### 2.4. Effect of HAMSB on Islet Function and Identity

To gain better insight into the mechanisms by which HAMSB improves glycemic control, we isolated islets and assessed glucose-stimulated insulin secretion. Basal and glucose-stimulated insulin release were similar in islets from Ctr and HAMSB-fed mice (Figure 5A), whereas islets from HAMS-fed mice secreted less insulin at 20 mM glucose (Figure 5B). When glucose-stimulated insulin secretion was augmented by forskolin, a non-significant increase was observed from islets isolated from HAMSB-fed mice compared to the other groups (*p* = 0.086) (Figure 5C). Total islet insulin content was significantly higher (36%) in islets from mice fed HAMSB compared to islets from Ctr mice (Figure 5D). Moreover, the expression of the insulin gene *Ins2* was 39% higher in islets from HAMSB-fed mice compared to Ctr mice (Figure 5F). The expression of other key beta-cell identity genes, such as *Ins1*, *MafA*, *Pdx1* and *Ucn3*, was not significantly different between the groups (Figure 5F). Total beta-cell area, determined as insulin-positive cells per pancreas area, did not differ between the groups (Figure 5E) and no obvious difference in the number of islets and islet size was found. Together, these results show that islets from the HAMSB-fed mice express higher levels of *Ins2* mRNA and contain more insulin compared to islets from control mice.

### 2.5. Effect of HAMSB on Lipid and Glucose Metabolism in the Liver

Because dysregulation of glucose and lipid metabolism in the liver promotes systemic metabolic dysfunction [41], we measured hepatic lipid accumulation and expression of genes involved in lipid and glucose metabolism. The levels of hepatic triglycerides were significantly lower (~9%) in HAMS- and HAMSB-fed mice compared to Ctr mice (Figure 6A). We analyzed the expression of genes involved in the regulation of lipid accumulation, such as uptake of fatty acids, de novo lipogenesis, fatty acid oxidation and/or export of fatty acids [42]. However, no differences were found in the expression of de novo lipogenesis genes *Acaca*, *Fasn*, *Scd1* and *Srebp-1c*, the fatty acid transporter *Cd36* and the rate-limiting enzyme in fatty acid oxidation *Cpt1a* (Figure 6B). The expression of *Pck1*, encoding phosphoenolpyruvate carboxykinase, and *G6p*, encoding glucose-6-phosphatase, was not statistically different between the groups (Figure 6C). Together, these results suggest that HAMS and HAMSB supplementation inhibit lipid accumulation in the liver, without affecting the expression of key genes involved in lipid metabolism.

### 2.6. Effect of HAMSB on Inflammation

Low-grade inflammation is associated with T2D and fatty liver disease [43]. To assess local inflammation, we measured the expression of genes associated with inflammation in the liver and adipose tissue. The expression of monocyte/macrophage marker genes *Cd68* and *F4/80* was significantly reduced in both the liver (Figure 7A) and adipose tissue of HAMSB-fed mice (Figure 7B). A non-significant reduction in the expression of the proinflammatory cytokine *Tnf-α* was observed in both liver and adipose tissue by HAMSB (Figure 7A,B). HAMS significantly reduced the expression of *Tnf-α* in the liver (Figure 7A), but not in the adipose tissue (Figure 7B). Systemic inflammation was assessed by measuring cytokine levels in plasma. However, no differences in neither proinflammatory cytokines (IL-1β, Tnf-α, IFN-ƴ, IL-2, IL-5, IL-6 and KC/Gro) nor the anti-inflammatory cytokine IL-10 were found between the groups (Appendix A). Together, these results indicate that HAMSB ameliorates inflammation locally in the liver and adipose tissue.

## 3. Discussion

In this study, we show that a diet containing butyrate in the form of HAMSB improves metabolic parameters associated with diabetes in db/db mice. HAMSB increased the insulin sensitivity, insulin content in pancreatic islets, and decreased the accumulation of hepatic triglycerides and inflammation in both the liver and adipose tissue. Together, these findings suggest anti-diabetic effects of HAMSB.

We used HAMS as a carrier to ensure targeted and sustained release of butyrate in the colon. Higher concentrations of butyrate were found in the feces of HAMSB-fed mice compared to HAMS-fed and control mice, indicating that this is an efficient way to increase the level of butyrate in the large intestine. Moreover, butyrate may decrease colonic pH, which can further boost the growth of butyrate-producing species [44,45]. As expected, the bacterial fermentation of HAMS promoted SCFA production. HAMSB also increased fecal acetate, suggesting that HAMSB modulates the overall gut microbiota community structure and promotes the growth of acetate-producing bacteria, or that butyrate is converted into acetate [46]. Therefore, we cannot rule out that the effects of HAMSB are solely caused by butyrate, as other metabolites could likely have an effect as well. However, it is important to note that fecal SCFAs only represent 5% of total SCFAs in the colon and will always reflect a balance between excretion, cross-feeding interactions and absorption [25,47]. Because butyrate is the main energy source for the colonocytes and is further metabolized in the liver, only low concentrations of butyrate can be measured in the circulation [46,48]. HAMSB has been shown to be superior to HAMS in increasing butyrate levels in the circulation [7,34]. We found that HAMSB tended to increase plasma levels of butyrate, but it was not different from HAMS. Possible explanations for this discrepancy could be the model used, duration of intervention, diet composition or measurement under fasted condition. Analysis of butyrate concentration in the portal vein might provide a more accurate measure of butyrate uptake. Unexpectedly, we also observed that HAMSB-fed mice gained more weight compared to controls, despite similar energy intake. This finding contradicts previous studies showing that butyrate inhibits weight gain in high-fat diet, obesity-induced animal models [15,16,19]. This could possibly be explained by increased energy harvest from the higher levels of butyrate, or it could be a result of cecum enlargement. Several studies have shown that fiber fermentation and SCFA production increase cecum tissue and content weight [15,16,19].

Key characteristics of T2D are insulin resistance and insufficient insulin production from beta-cells, leading to hyperglycemia. We observed that islets of mice fed HAMSB contained more insulin, had higher insulin gene expression, and tended to secrete more insulin in response to stimulation with 20 mM glucose and forskolin compared to controls. Together, this suggests that islets of HAMSB-fed mice have a higher capacity to secrete insulin under increased metabolic demand. The db/db mice have a mutation in the leptin receptor, and display hyperphagia, and consequently develop obesity, hyperglycemia and insulin resistance, resembling key features of human T2D [40]. The beta-cells adapt to these metabolic challenges by producing and secreting more insulin to prevent hyperglycemia [49]. Although the mice fed HAMSB also developed hyperglycemia, these mice had decreased HOMA-IR, indicating improved insulin sensitivity compared to the control mice. This might explain the reduction in integrated blood glucose over the 5-week treatment period observed in HAMSB-fed mice. No effect of HAMSB feeding was observed on the beta-cell area as percent of total pancreas area, indicating no effect on beta-cell growth and apoptosis. However, these data were obtained using relatively few mice and total beta-cell mass, proliferation and apoptosis were not measured in this study. Whether the effects of butyrate on the islets are direct or secondary to its action on other tissues remains unknown. However, in the present study, we show that butyrate improves beta-cell function in islets of db/db mice in vitro. This, together with our previous findings that butyrate protects beta-cells from cytokine-induced dysfunction [38], indicates direct beneficial effects of butyrate on beta-cell function. It can be speculated whether the concentration of butyrate in the plasma of the db/db is sufficient to elicit a direct response in beta-cells. If not, the effects of the HAMSB on the beta-cells must be indirect. This could be explained by butyrate reducing blood glucose, local tissue inflammation, hepatic lipid accumulation and improved insulin sensitivity, which ameliorate stress from the beta-cells and could thereby, indirectly, maintain insulin production and reserve.

The liver plays a key role in lipid and glucose metabolism [50]. We observed that mice fed HAMSB had reduced lipid accumulation and inflammatory markers in the liver compared to controls. Similar beneficial effects of butyrate have previously been described in experimental models using butyrate mixed into the diet [16,51,52]. Hepatic steatosis impairs liver function and develops as a consequence of an imbalance between de novo lipogenesis, fatty acid oxidation and import and export of lipids [42]. We did not find any changes in the expression of selected genes involved in these processes, but other genes or posttranslational events could be important. Moreover, failure of insulin to inhibit lipolysis in the adipose tissue would release more fatty acids into the circulation, resulting in higher uptake in the liver [53]. Kosteli and coworkers showed that lipolysis in the adipose tissue recruits macrophages [54]. Notably, adipose tissue inflammation was reduced in mice fed HAMSB, suggesting that lipolysis is also reduced, and thus this could be a potential mechanism by which butyrate inhibits hepatic steatosis. 

The gut is connected to the liver via the portal vein, and therefore the concentration of gut metabolites, such as butyrate, is higher in the liver compared to the periphery [55]. We have previously shown that HAMSB increases the levels of butyrate in portal blood [7]. Both HAMS and HAMSB had beneficial effects on the liver, indicating that these effects might result from the HAMS carrier. It is likely, that the concentration of butyrate or any of the other SCFAs delivered from HAMS or HAMSB are sufficient to directly affect the hepatocytes, e.g., via signaling through G protein-coupled rectors or inhibition of histone deacetylases. Direct effects of butyrate in the liver have been reported by others [56,57,58].

Since we deliver butyrate targeted to the large intestine, some of the effects of HAMSB (and HAMS) might also be directly mediated via the gut microbiota. Others have shown that butyrate in the gut stimulates the secretion of the gut hormones glucagon-like peptide (GLP-1) and peptide YY (PYY) from the enteroendocrine L-cells that play an important role in decreasing blood glucose by promoting insulin secretion and satiety [59,60,61]. Butyrate also improves the intestinal barrier function by increasing tight junction assembly [16] and mucus production [62]. Hyperglycemia and gut microbiota dysbiosis drive intestinal barrier dysfunction, allowing the passage of bacterial components and dietary antigens that can trigger an inflammatory response [63,64]. Improvement of the barrier function by butyrate could, thus, decrease inflammation locally and systemically. Accordingly, HAMSB reduced the expression of proinflammatory cytokine, *Tnf-α*, and macrophage markers, *Cd68* and *F4/80*, in the liver and adipose tissue. Direct anti-inflammatory activity of butyrate via an inhibition of NF-κB signaling has also been reported by us and others [39,65,66,67]. As inflammation has been associated with metabolic dysfunction, the anti-inflammatory effect of HAMSB, whether through an improvement of the intestinal barrier and/or directly, suggests a potential mechanism by which butyrate improves insulin sensitivity, glycemic control and hepatic lipid accumulation. These findings are in line with previous studies showing the immunoregulatory properties of HAMS esterified to SCFAs [7,8].

We acknowledge that the db/db model of T2D has limited translational potential, as T2D in human is multifactorial and not caused by a single gene mutation. The db/db mouse model is a severe model and mice display hyperglycemia from a young age [40,68]. Recently, Li et al. showed anti-diabetic effects of different SCFAs bound to HAMS in a high-fat diet and streptozotocin (STZ)-induced rat model of T2D [34]. However, in this mode, l beta-cells were rapidly destroyed after STZ injection, and did not replicate the slow progression of beta-cell dysfunction as seen in human T2D [69]. On the other hand, the db/db model is a severe model of T2D and this could have prevented us from observing beneficial effects of HAMSB that could be relevant in an earlier pre-diabetic stage. Therefore, future studies need to address the potential of HAMSB in less severe models of T2D, such as high-fat, diet-induced models. Collectively, we demonstrate that HAMSB supplementation has beneficial effects on insulin sensitivity, insulin synthesis, and hepatic lipid accumulation, possibly via inhibition of local inflammation. Interestingly, a recent human study observed that enhanced SCFA levels and immunoregulatory effects of HAMSAB persisted 6 weeks after treatment intervention [8], showing that the effects are not transient. Together, this suggests that targeted delivery of butyrate, with HAMS as a carrier, may have potential in prevention and treatment of T2D.

## 4. Materials and Methods

### 4.1. Ex Vivo Stimulation of Islets of db/db Mice

Pancreatic islets from 9- and 13-week old male db/db mice (BKS.Cg -+Lepr^db^/+Lepr^db^/OlaHsd, Envigo, Horst, The Netherlands) fed a chow diet were isolated, as previously described [39]. The day after isolation, glucose-stimulated insulin secretion was performed on 25 islets in duplicate, and the rest of the islets were cultured in the RPMI 1640 medium (Gibco, Thermo Fisher Scientific, Roskilde, Denmark), supplemented with 2% human serum (BioWhittaker, Basel, Switzerland), and 1% penicillin (100 U/mL) and streptomycin (100 μg/mL) (P/S) for 10 days, with and without 0.2 mM butyrate (B5887, Sigma, Soeborg, Denmark). On day 5, the islets were transferred to new media with and without butyrate. On day 10, GSIS was performed, as described below. To compare diabetic vs. non-diabetic islets the day after isolation, islets from 11-week old mice (*C57BL*/6NRj, Janvier, Saint Berthewin Cedex, France) were included.

### 4.2. Experimental Design of Mouse Study

Five-week old male db/db mice (BKS(D)-Leprdb/JOrlRj) were purchased from Janvier. The mice were acclimatized to the animal facility at 20–22 °C with a 12 h light/dark cycle for one and a half weeks prior to the start of the study. The mice were then randomized into three dietary groups based on body weight and blood glucose measurements, to ensure similar baseline characteristics between the groups. The mice were fed either: (1) a control diet (AIN-93G, Ctr), (2) AIN-93G with 15% corn starch replaced by HAMS (HAMS), or (3) AIN-93G with 15% corn starch replaced by butyrylated HAMS (HAMSB) for 5 weeks. The diets were formulated by Envigo and the composition is found in Appendix A. The mice were housed in groups of three to four per cage and had free access to food and water. Food intake, body weight and fasting blood glucose (4 h) were measured once a week. Blood was taken from the tail-tip and glucose levels measured using a glucometer (Bayer Contour, Leverkusen, Germany). After 5 weeks on the diet, the mice were fasted (4 h) before termination of the experiment. The mice were anesthetized using isoflurane and blood was collected by cardiac puncture, after which they were sacrificed by cervical dislocation. Liver and epididymal fat pads were collected and immediately snap-frozen and stored at −80 °C until further analyses. Pancreases from three mice per group were fixed in 4% formaldehyde and pancreatic islets were isolated as previously described from remaining mice [39]. The experiments were approved by the local ethics committee, and animals were housed according to the Principles of Good Laboratory Care. Mice with a fasting blood glucose above 24 mM in week 1 were excluded from the analysis (three mice, from both the Ctr and HAMS group, and two from the HAMSB group).

### 4.3. Short-Chain Fatty Acid Analyses

SCFA concentrations in fecal and plasma samples were determined by liquid chromatography-tandem mass spectrometry. To prepare the fecal sample, they were mixed with 20 µL internal standard mix (1 mM D3-acetate, 5 mM D4-propionate, 0.1 mM D7-butyrate) and freeze-dried, weighted and homogenized by sonication in Milli-Q water. For both fecal and plasma samples, the derivatizing reagent was 200 mM 3-Nitrophenylhydrazine (NPH) and 120 mM N-(3-Dimethylaminopropyl)-N′-ethylcarbodiimide hydrochloride (EDC) in 50% acetonitrile (AcN) (with 6% pyridine). Freeze-dried fecal samples (20 µL) were mixed with derivatizing reagent (40 μL) and incubated for 1 h at room temperature. Plasma samples (10 µL) were mixed with 10 μL of internal standard mix in 50% methanol (0.1 mM D3-acetate, 1 mM D4-propionate, 0.1 mM D7-butyrate) and derivatizing reagent (40 μL), and incubated for 1 h at room temperature. All the samples were centrifuged at 14,000 rpm for 10 min at 4 °C, and mixed 1:1 with 10% AcN. The quantification of the SCFAs was achieved on the liquid chromatography-tandem mass spectrometer consisting of a Waters Acquity UPLC I-Class connected to a Waters Xevo TQ-XS tandem mass spectrometer (Waters, Manchester, UK). The separation was performed by injecting 2 µL of each sample to an Acquity UPLC HSS-T3 column (100 × 2.1 mm, 1.8µm, Waters, MA, USA). The mobile phase consisted of 0.1% formic acid and AcN with 0.1% formic acid, and was delivered on the column with a flow rate of 0.40 mL/min with the following gradient: 3% B to 55% B in 7 min, and then up to 100% for 1 min, hold for 2 min, and thereafter back to initial condition, in 0.5 min and equilibrated for 2.5 min. Column and autosampler were thermostated at 55 °C and 4 °C, respectively. Analytes were ionized in an electrospray ion source operated in the negative mode. The source and gas parameters were set as follows: ion spray voltage 3.5 kV, desolvation temperature 300 °C, desolvation gas flow 800 L/h, nebulizer pressure 7 Bar, and cone gas flow 150 L/h. The instrument was operated in multiple reaction monitoring mode (MRM). Dwell time was set to 35 ms for all transitions. For quantification 6-point calibration curves were used, including different levels of non-labelled and constant levels of the labelled internal standards. The instrument was controlled by MassLynx 4.2, and data processing was performed with TargetLynx XS (Waters, Milford, MA, USA).

### 4.4. Blood Samples

Terminal blood was collected by cardiac puncture in EDTA tubes and placed immediately on ice, after which they were centrifuged at 5000× *g* for 10 min at 4 °C. Plasma was collected and stored at −80 °C until analysis. Plasma levels of insulin were measured using a mouse insulin ELISA kit (80-INSMS-E01, Alpco, Salem, NH, USA), according to the manufacturer’s instruction. As an index of insulin resistance, HOMA-IR was calculated (HOMA-IR = fasting glucose (mmol/L) × fasting insulin (μU/mL)/22.5). Proinflammatory cytokine levels were determined using the V-plex proinflammatory Panel 1 mouse kit (Meso Scale Discovery, # K15048D, Rockville, MD, USA), according to the manufacturer’s instructions, using a MESO QuickPlex SQ 120 instrument.

### 4.5. Glucose-Stimulated Insulin Secretion

After overnight culture in RPMI 1640 media with 1% P/S and 10% fetal bovine serum (FBS, Biosera, Herlev, Denmark), islets were collected in groups of 25 islets of similar size in duplicate from each mouse, and preincubated in 24-well plates with the Krebs–Ringer HEPES (KRBH) buffer, supplemented with 2 mM glucose for 1.5 h at 37 °C. The islets were then transferred to new wells with 2 mM glucose in the KRBH buffer for 30 min, half of the buffer was collected and the concentration of the glucose was increased to 20 mM. After 30 min, half of the buffer was collected and forskolin (20 µM) was added for 30 min. Then, the buffer was collected, and the islets were saved for the determination of cellular insulin and DNA content after sonication of the samples. Insulin was quantified using an in-house developed insulin ELISA. Insulin secretion was normalized to DNA content, determined by the Quant-IT PicoGreen dsDNA Reagent and Kit (Thermo Fisher Scientific, Roskilde, Denmark). The KRBH buffer contained: 115 mM NaCl, 10 mM HEPES, 5 mM NaHCO_3_, 4.7 mM KCl, 2.6 mM CaCl_2_, 1.2 mM KH_2_PO_4_, 1.2 mM MgSO_4_, 0.2% BSA, 2 mM glutamine, and 1% P/S, pH 7.4.

### 4.6. RNA Isolation and Gene Expression

Liver (20 mg) and adipose tissues (100 mg) were homogenized in TRIzol reagent with Qiagen Tissuelyser (2 × 2 min, 20 Hz). Adipose tissue samples were centrifuged at 12,000× *g* for 10 min at 4 °C, to allow a triglyceride phase to form and the aqueous phase was collected. Phase separation was performed using chloroform, followed by centrifugation at 12,000× *g* for 15 min at 4 °C. For RNA isolation from islets, pools of 300–500 islets from 1–5 mice were lyzed in TRIzol reagent. For all samples, total RNA was extracted using the Direct-zol RNA-mini prep kit (Zymo Research, Nordic Biosite), according to the manufacturer’s protocol. RNA quality and quantity was assed using Nanodrop, and cDNA was synthesized using qScript cDNA Super mix kit (Quantabio). Gene expression in islets was determined using TaqMan, whereas gene expression in the liver and adipose tissue was determined using SYBR. Probes and primers are found in Appendix A. RT-QPCR was performed on an ABI PRISM 7900HT Sequence Detection System (Applied Biosystems). Each sample was determined in triplicate and the expression of target genes was normalized to the expression of *Ppia* (encoding cyclophilin A).

### 4.7. Immunohistochemistry and Beta-Cell Area

The fixated pancreases were dehydrated in ethanol, cleared in xylene and embedded in paraffin. The paraffin blocks were cut in 4 µm sections, using a microtome. The sections were transferred to slides, dewaxed in Histolab-Clear (Histolab Products AB, #14250) and rehydrated in graded dilutions of alcohol (70–99%) in tap water. The slides were stained for insulin using a primary in-house antibody (Insulin 2006-4). The signal was amplified by a biotinylated secondary antibody, goat anti-guinea pig (BA-7000, Vector Laboratories, Newark, CA, USA), followed by peroxidase (ABC) (PK-4000, Vector Laboratories). Finally, the reaction was developed by the use of 3,3–diaminobenzidine (SK-4100, Vector Laboratories), and counterstaining was performed with a Hematoxylin Mayer solution (RH-pharmacy, #854183). The slides were examined and photographed using the Zeiss Axio Scanner Z1. The islet area (percent insulin-positive tissue relative to total tissue area) and insulin intensity were quantified using the QuPath software (version0.3.2) Pancreases from three mice per group were analyzed. Three sections per mice were quantified and the average was calculated.

### 4.8. Hepatic Triglycerides

Triglycerides in the liver were quantified using a colorimetric triglyceride assay kit (Ab65336, Abcam, Cambridge, UK), according to the manufacturer’s instructions. Briefly, 60 mg liver tissue was homogenized in 600 µL 5% NP-40 in water, using a pestle. The samples were heated to 90 °C for 5 min and cooled down to room temperature. This was repeated twice to solubilize all triglycerides, and the samples were centrifuged at maximum speed to remove insoluble material. The supernatant was collected, diluted 10-fold in water, and 4 µL was assayed. Absorbance was determined at 570 nm.

### 4.9. Statistics

Statistical analyses and graphs were performed using the GraphPad Prism software Version 9.3.1. For longitudinal measures of blood glucose and body weight, data were analyzed by a repeated measures ANOVA, followed by a Tukey’s test. For comparisons of the two groups, a two-tailed unpaired Student’s t-test was performed if data were normally distributed, otherwise a non-parametric Mann–Whitney u-test was performed. Outliers were defined as values above or below the mean +/− standard deviations, and were excluded from the data. Data are represented as the mean +/− SEM, and a *p*-value < 0.05 was considered statistically significant. Each symbol represents an individual mouse, unless otherwise specified. For blood glucose measurements over time, AUCs of blood glucose measurements were calculated using the incremental AUC.

## Figures and Tables

**Figure 1 ijms-24-04533-f001:**
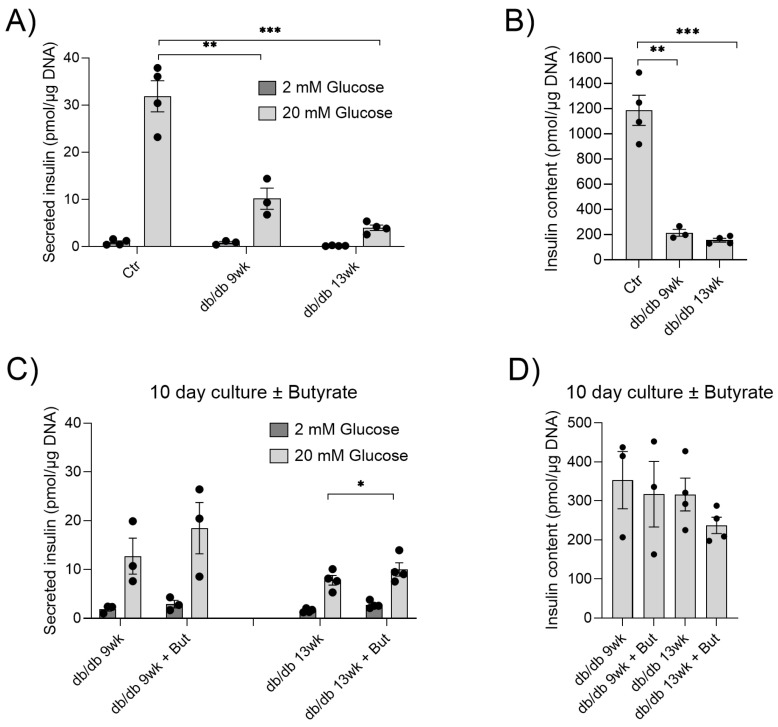
In vitro effects of butyrate on glucose-stimulated insulin secretion and insulin content in islets. (**A**) Insulin secretion from islets isolated from control C57BL/6 mice (Ctr), or db/db mice, at 9 weeks (9 wk) or 13 weeks (13 wk) of age, was measured by static batch incubations in response to 2 mM and 20 mM glucose, and normalized to DNA. (**B**) Insulin content measured post-assay. (**C**) Insulin secretion from db/db islets cultured with and without 0.2 mM butyrate (But) for 10 days. (**D**) Insulin content measured post-assay. Bars show the mean ± SEM. * *p* < 0.05, ** *p* < 0.01 and *** *p* < 0.001.

**Figure 2 ijms-24-04533-f002:**
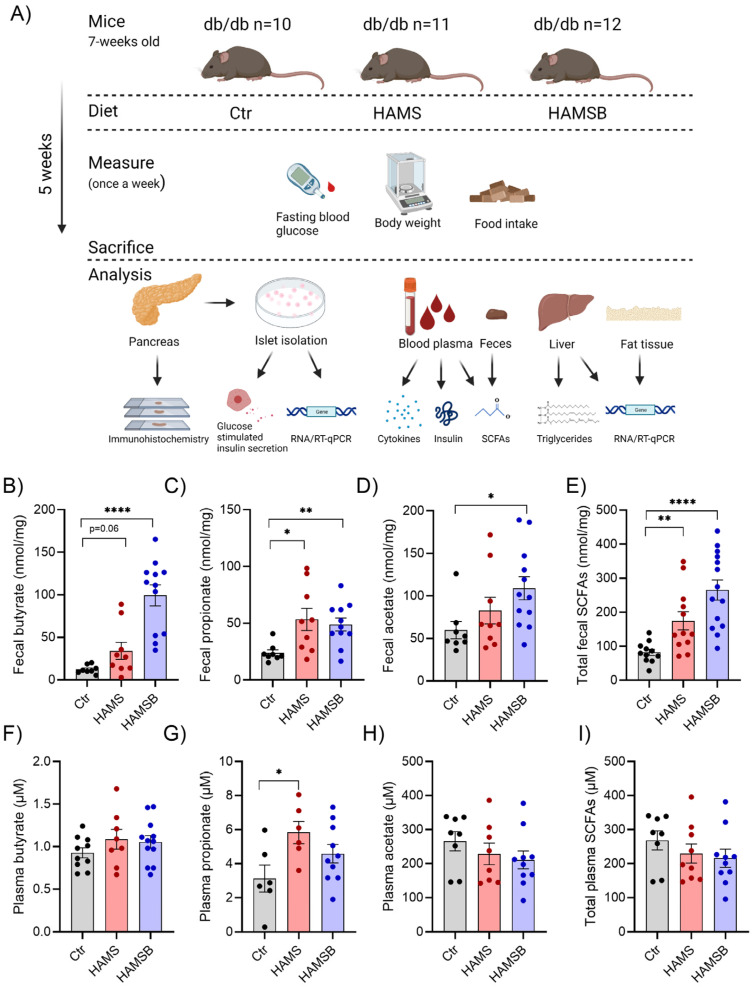
Experimental overview and SCFA concentrations in feces and plasma. (**A**) Db/db mice were fed either a control diet (Ctr, AIN-93G), a diet supplemented with 15% HAMS, or a diet supplemented with 15% HAMSB for 5 weeks. Blood glucose, body weight and food intake were measured weekly. After 5 weeks, the mice were sacrificed, and tissues were collected for analyses. SCFA concentrations were measured in the feces (**B**–**E**) and plasma (**F**–**I**) of db/db mice at the end of the study. Bars show the mean ± SEM. * *p* < 0.05, ** *p* < 0.01 and **** *p* < 0.0001.

**Figure 3 ijms-24-04533-f003:**
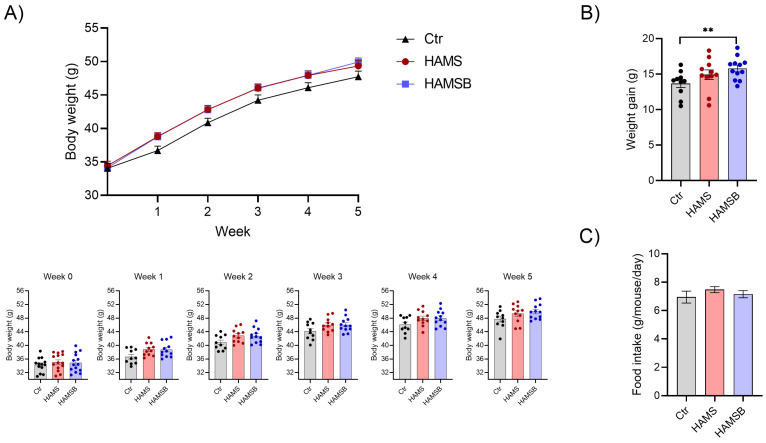
Bodyweight and food intake. (**A**) Weekly measurements of body weight in db/db mice fed a control diet or a diet supplemented with either HAMS or HAMSB. Body weight for each individual mouse in the different weeks are shown below the weight curve. (**B**) Total weight gain at the end of the 5-week feeding period. (**C**) Food intake is shown as gram per mouse per day. Bars show the mean ± SEM. ** *p* < 0.01. Each circle represents an individual mouse.

**Figure 4 ijms-24-04533-f004:**
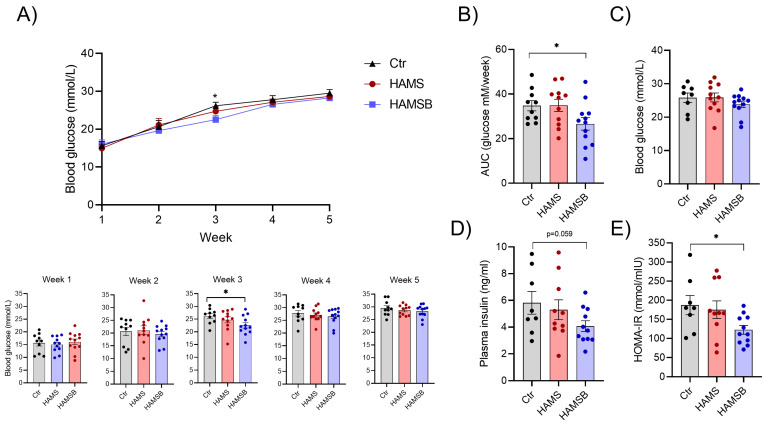
Glycemic control. (**A**) Weekly measurements of fasting blood glucose in db/db mice fed three different diets: Ctr, HAMS or HAMSB for 5 weeks. Blood glucose levels for each individual mouse in the different weeks are shown below the blood glucose curve. (**B**) Area under the blood glucose curve (AUC) was calculated for each mouse. On the day of termination after a 4 h fast, blood glucose (**C**) and plasma insulin (**D**) were measured and used to determine insulin sensitivity through HOMA-IR (**E**). Bars show the mean ± SEM. * *p* < 0.05. Each circle represents an individual mouse.

**Figure 5 ijms-24-04533-f005:**
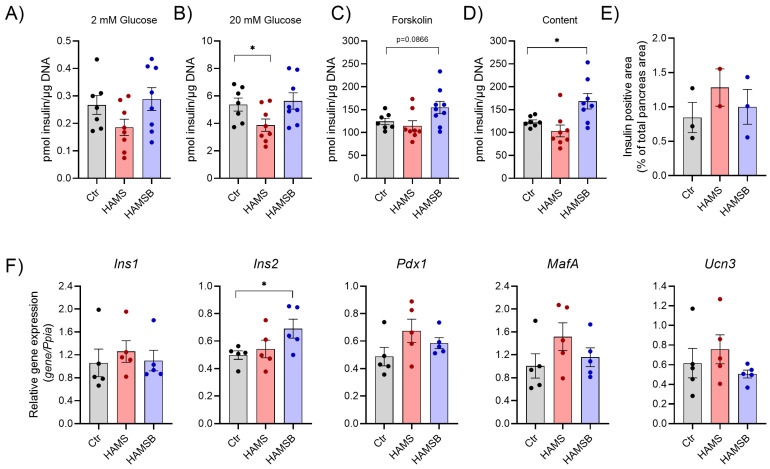
Islet function and gene expression. Insulin secretion was measured from islets of db/db mice fed three different diets: Ctr, HAMS or HAMSB for 5 weeks. The day after islet isolation, insulin secretion was measured by static batch incubations in response to 2 mM glucose (**A**), 20 mM glucose (**B**) and 20 µM forskolin with 20 mM glucose, and normalized to DNA (**C**). (**D**) Insulin content measured post-assay. (**E**) Beta-cell area was detected by immunohistochemistry and shown as the insulin-positive area, as a percentage of the total pancreas area. (**F**) Gene expression was determined by RT-qPCR in pancreatic islets cultured overnight. Total RNA was extracted from a pool of islets from 1–5 mice on the same diet. Relative mRNA abundances of the insulin genes: *insulin 1* (*Ins1*) and *insulin 2* (*Ins2*) and beta-cell identity genes: *pancreatic and duodenal homeobox 1* (*Pdx1*), *MAF bZIP transcription factor A* (*MafA*) and *urocortin3* (*Ucn3*) are shown. Data are normalized to the reference gene *peptidylprolyl isomerase A* (*Ppia*). Bars show the mean ± SEM. * *p* < 0.05. Each circle represents pools of islets from mice.

**Figure 6 ijms-24-04533-f006:**
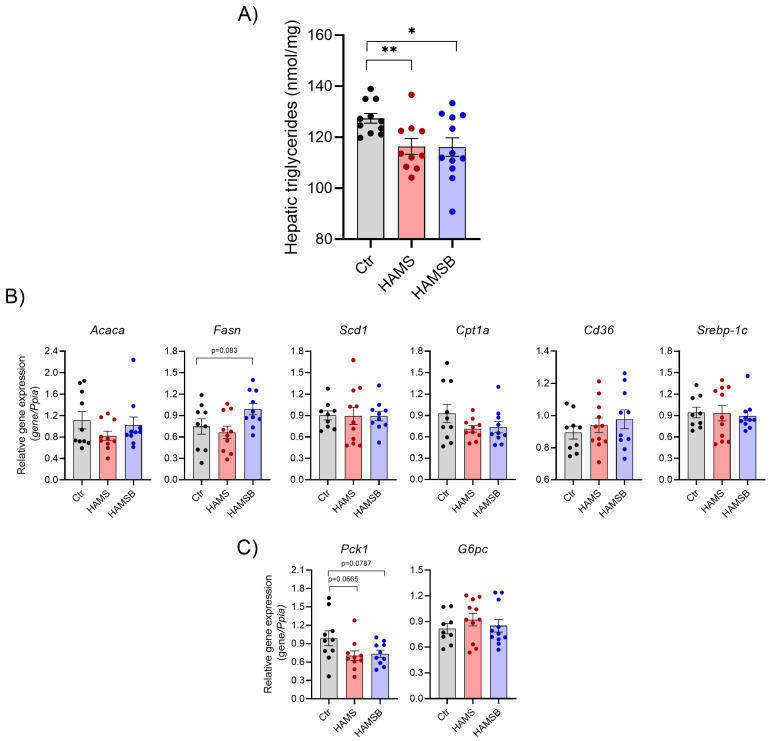
Hepatic lipid and glucose metabolism. (**A**) Triglycerides were measured in livers of db/db mice fed a control diet or a diet supplemented with HAMS or HAMSB for 5 weeks. Total RNA was extracted from the livers and gene expression of the genes involved in lipid metabolism (**B**) or gluconeogenesis (**C**) were measured by RT-qPCR. Data are normalized to the reference gene *Ppia*. Bars show the mean ± SEM. * *p* < 0.05, ** *p* < 0.01. *Acaca*, *acetyl-CoA carboxylase; Fasn*, *fatty acid synthase*; *Scd1*, *stearoyl-CoA desaturase 1*; *Cpt1a*, *carnitine palmitoyltransferase 1A*; *Srebp-1c*, *sterol regulatory element-binding protein-1c*; *Cd36*, *cluster of differentiation 36*; *Pck1*, *phosphoenolpyruvate carboxykinase 1*; *G6pc*, *glucose*-*6*-*phosphatase.* Each circle represents pools of islets from mice.

**Figure 7 ijms-24-04533-f007:**
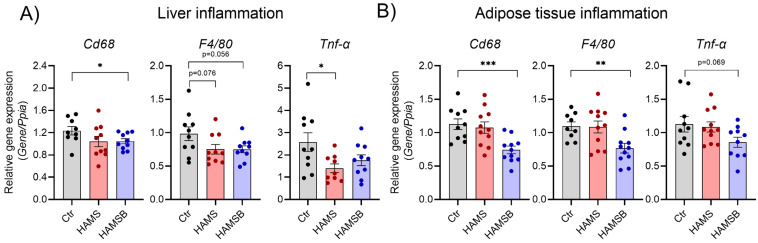
Inflammation in hepatic and adipose tissue. Inflammatory gene expression in the livers (**A**) and epididymal adipose tissue (**B**) of mice fed a Ctr, HAMS or HAMB diet for 5 weeks. Total RNA was extracted, and gene expression determined by RT-qPCR. Data are normalized to the reference gene *Ppia*. Bars show the mean ± SEM. * *p* < 0.05, ** *p* < 0.01, *** *p* < 0.001. *Cd68*, *cluster of differentiation 68*; *F4/80* encoded by *Adgre1*, *adhesion G protein-coupled receptor E1*; *Tnf-α*, *tumor necrosis factor α.* Each circle represents pools of islets from mice.

## Data Availability

No supporting data to this article are available.

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
