# Peer review of "Targeted Delivery of Butyrate Improves Glucose Homeostasis, Reduces Hepatic Lipid Accumulation and Inflammation in db/db Mice"

_ijms, 2023, doi:10.3390/ijms24054533_

Round 1
Reviewer 1 Report
Dear editor,
Please find the review comments,
Dr.BUSA.

Author Response
Thank you for providing a review of our manuscript. We have made major revisions to the manuscript and believe that the revised version is significantly improved. The individual points have been addressed as described below:
1. The entire abstract has been rewritten in a more scientific style. We hope the reviewer will find this version acceptable.
2. The introduction has been revised substantially. The individual changes can be seen in the track changes version of the revised manuscript. In the introduction we systematically introduce beta cell dysfunction as a major factor in the development of type 2 diabetes, this is followed by a description of the role of the gut microbiota in metabolic health with focus on short chain fatty acids. Effects of butyrate in other inflammatory condition is then introduced and finally we explain the rationale for the present study. We hope the reviewer agree that this revised version of the introduction is a fair and informative introduction to the study.
3. In the Materials and Methods section, we have described every method used in the generation of the data presented in the manuscript. We believe we have provided sufficient information for others to replicate the results as required by IJMS “instructions to authors”. Specifically, we have described the procedure for “Histopathological examination of the pancreas”. We believe the description of this procedure to be accurate of sufficient detail for other to replicate this. The analysis was performed in order to analyze the beta cell area relative to total pancreas area. This is an important parameter when addressing the compensation of beta cell growth in response to insulin resistance as seen in db/db mice. The data from this analysis is shown in Figure 5E.
4. We have not included a “Conclusion” section to our manuscript. It is clearly stated in the IJMS instructions to authors (https://www.mdpi.com/journal/ijms/instructions) that “Conclusions: This section is not mandatory but can be added to the manuscript if the discussion is unusually long or complex”. Therefore, we do not think that we have committed any error in terms of complying with the formal requirements for manuscripts submitted to IJMS.
Reviewer 2 Report
In the present study, the authors investigated the effects of targeted delivery of butyrate using specially designed prebiotic starch on glucose metabolism, beta-cell function, and hepatic lipid accumulation. Overall, the manuscript is well-written and sufficiently detailed. Below are some suggestions for improving the manuscript.
1. Beta cell mass should be calculated by multiplying relative insulin-positive area by pancreas weight. However, in the manuscript only measurement of relative insulin-positive area was shown. Can the authors provide beta cell mass calculation? Another issue is that the number of animals included in insulin-positive area measurement was quite limited.
2. How did butyrate treatment affect beta-cell apoptosis and proliferation in this study?
3. In the Results section, some of the statistical analyses reported in the figures are not statistically significant, such as Figure 4D, Figure 5C, Figure 6C, and Figure 7. The authors should be more careful about the language used for describing those results.
Author Response
Thank you for reviewing our manuscript. We have read the comments carefully and have made substantial revision based on the comments.
- We certainly agree that a measure of beta-cell mass would be relevant and superior to the data on relative insulin-positive area presented in the manuscript. As mentioned by the reviewer, analysis of beta cell mass would require a valid measurement of total pancreatic weight. In the db/db mice massive fat accumulation is present in the abdomen and it is not possible to dissect the pancreatic tissue without significant “contamination” with adipose tissues. In fact, substantial amount of adipose tissue is present within the pancreas of these mice making it impossible to measure true pancreatic weight. We realize that the number of animals in each group (n=3 for controls and HAMSB-fed mice) is low. In order to comply with 3R principles in animal research we used the minimal number of mice for this project. With a focus on islet function we favored the analysis of isolated islets over histology and thus were only able to include 3 animals for histology. Discussion of these limitations has now been included in the discussion section of the manuscript on page 10.
- We agree that analysis of beta cell apoptosis and proliferation would be interesting. However, in this study we did not observe any effects of HAMSB feeding on relative beta-cell area and we did find it necessary to include the very complex analysis of beta-cell apoptosis and proliferation. These limitations of our study are now included in the discussion on page 10.
- We have left the p-values for the statistical analysis of the various differences in Figures 4D, 5C, 6BC and 7. We find it relevant to provide this information to the reader when p-values are between 0.1 and 0.05. However, we have changed the description of the data in these Figures, and we now clearly state in the result section that these differences are non-significant, when the p<0.05 criteria is used, and the overall description of the data reflects this.
Round 2
Reviewer 1 Report
Dear editor,
The manuscript has been sufficiently improved by authors, please accept in the current revised form.
Dr.BUSA.